# Assessment of health system readiness for routine maternal and newborn health services in Nepal: Analysis of a nationally representative health facility survey, 2015

Resham B. Khatri[1,2]*, Yibeltal Assefa[1], Jo Durham[1,3]

1 School of Public Health, Faculty of Medicine, University of Queensland, Brisbane, Australia, 2 Health Social Science and Development Research Institute, Kathmandu, Nepal, 3 School of Public Health and Social Work, Queensland University of Technology, Brisbane, Australia

* rkchettri@gmail.com

**Data Availability Statement:** Yes - all data are fully available without restriction. Data used in this study

## Abstract

Access to and utilisation of routine maternal and newborn health (MNH) services, such as antenatal care (ANC), and perinatal services, has increased over the last two decades in Nepal. The availability, delivery, and utilisation of quality health services during routine MNH visits can significantly impact the survival of mothers and newborns. Capacity of health facility is critical for the delivery of quality health services. However, little is known about health system readiness (structural quality) of health facilities for routine MNH services and associated determinants in Nepal. Data were derived from the Nepal Health Facility Survey (NHFS) 2015. Total of 901 health facilities were assessed for structural quality of ANC services, and 454 health facilities were assessed for perinatal services. Adapting the World Health Organization's Service Availability and Readiness Assessment manual, we estimated structural quality scores of health facilities for MNH services based on the availability and readiness of related subdomain-specific items. Several health facility-level characteristics were considered as independent variables. Logistic regression analyses were conducted, and the odds ratio (OR) was reported with 95% confidence intervals (CIs). The significance level was set at p-value of <0.05. The mean score of the structural quality of health facilities for ANC, and perinatal services was 0.62, and 0.67, respectively. The average score for the availability of staff (e.g., training) and guidelines-related items in health facilities was the lowest (0.37) compared to other four subdomains. The odds of optimal structural quality of health facilities for ANC services were higher in private health facilities (adjusted odds ratio (aOR) = 2.65, 95% CI: 1.48, 4.74), and health facilities supervised by higher authority (aOR = 1.96; CI: 1.22, 3.13) while peripheral health facilities had lower odds (aOR = 0.13; CI: 0.09, 0.18) compared to their reference groups. Private facilities were more likely (aOR = 1.69; CI:1.25, 3.40) to have optimal structural quality for perinatal services. Health facilities of Karnali (aOR = 0.29; CI: 0.09, 0.99) and peripheral areas had less likelihood (aOR = 0.16; CI: 0.10, 0.27) to have optimal structural quality for perinatal services. Provincial and local governments should focus on improving the health system readiness in peripheral and public facilities to deliver quality MNH services. Provision of trained staff and

are publicly available secondary data of Nepal Health Facility Survey (NHFS) 2015 obtained from the Demographic and Health Surveys (DHSO Program (https://dhsprogram.com/data/available-datasets.cfm) program. The NHFS 2015 was the first health facility survey conducted under the leadership of Ministry of Health and Population (Nepal) and technical support ICF Marco International, Maryland, USA, and New Era (local implementing partner in Nepal). Like any other DHS dataset, the NHFS 2015 dataset are publicly available for research and further analysis, any interested researchers can request those data from the DHS program and use for their research and analysis.

**Funding:** The authors received no specific funding for this work.

**Competing interests:** The authors have declared that no competing interests exist.

guidelines, and supply of laboratory equipment in health facilities could potentially equip facilities for optimal quality health services delivery. In addition, supervision of health staff and facilities and onsite coaching at peripheral areas from higher-level authorities could improve the health management functions and technical capacity for delivering quality MNH services. Local governments can prioritise inputs, including providing a trained workforce, supplying equipment for laboratory services, and essential medicine to improve the quality of MNH services in their catchment.

## Introduction

The health of mother and newborns from conception to postnatal is commonly referred to as maternal and newborn health (MNH). Uptake of health services during antenatal period (conception to before childbirth), and perinatal (28 weeks after conception to the first week after childbirth) is vital for improved health status of mothers and newborns [1]. Inadequate access to quality ANC and perinatal care contribute significantly to preventing several maternal and newborn deaths. Furthermore, high-quality ANC presents a unique and lifesaving opportunity for health promotion, disease prevention, early diagnosis and treatment of illnesses in pregnancy using evidence-based practices. To ensure optimum care, the World Health Organisation (WHO) recommended that every pregnant woman have a minimum of four ANC visits throughout the pregnancy, with the first visit in the first trimester [2]. Furthermore, routine ANC visits and childbirth in health facilities assisted by skilled birth attendants ensure antenatal, intrapartum care and immediate maternal and newborn care, and reduce the risk of adverse pregnancy outcomes, including perinatal morbidities and mortalities [3–5].

In the last two decades, Nepal has made significant progress in access to routine MNH visits such as at least four antenatal care visits (4ANC), institutional delivery, and at least one postnatal care (PNC) visit within 48 hours of childbirth. For instance, the uptake of institutional delivery increased from 3% in 1996 to 57% in 2016, and similar patterns of increment were observed in the 4ANC, and PNC visits [6, 7]. However, this increased access to routine services has not been reflected in MNH outcomes. For example, from 1996 to 2006, Maternal Mortality Ratio (MMR) reduced from 539 (reported as per 100000 live births) to 281, and Neonatal Mortality Rate (NMR) reduced from 50 (reported as per 1000 live births) to 33. But in the subsequent decade (2007–2016), MMR reduced from 281 to 259, and NMR reduced 33 to 22 only [6].

Evidence suggests that the reasons for slow progress in MNH outcomes are partly contributed by equity gaps in access to services, and utilisation of poor-quality health services. For instance, in 2016, access to institutional delivery among women of the lowest and highest wealth quintile was 34% and 90%, respectively [6]. Women with multiple forms of disadvantage had the lowest coverage of all MNH visits compared to their privileged counterparts [8]. Socioeconomically disadvantaged women from remote areas of Karnali province face access barriers to reach health facilities, which are further compounded by poor transportation systems [9–12].

Furthermore, women who attended health facilities received poor quality MNH services, especially facilities in peripheral areas [13]. Good quality health services need better health system inputs such as the provision of trained and technically competent health workforces, regular supply of essential medicine, and enabling environment (e.g., infrastructure, equipment). Such health system inputs determine the health facility readiness for quality health services and are the precondition for delivering quality health services [14].

The measurement of health care quality is complex, multifaceted, and depends on context, it also requires multiple data on health system inputs and processes of health services delivery. According to the Donabedian model, health care quality comprises three components: structural (inputs), process, and outcomes [15]. Structure denotes the attributes of the settings in which care occurs. Structural quality is health facility capacity to deliver good quality health services [16]. Process quality is the delivery of good quality technical services [17]. Process denotes what is done in giving and receiving care. Good structural quality usually depends on inputs in the health system and leads to the process of care or delivery of good quality health services. Outcome denotes the effects of care on the health status of patients and populations. The outcome component of quality refers to client satisfaction or improved health status of people [18].

Global health policies, plans and strategies, evidence on quality of care of maternal and child health services [4, 19–23], and focus on universal access to quality health services to achieve health-related Sustainable Development Goals (SDG3). Recent health policies of Nepal such as the Nepal Health Sector Strategy (2016–2021) [24], Strategy for Skilled Health Personnel and Skilled Birth Attendants 2020–2025 [25], Nepal Safe Motherhood and Newborn Health Road Map 2030 [26], and Nepal Newborn Action Plan (2015–2035) prioritise the quality of care for improved health outcomes. These policies have envisioned optimal health facility readiness, delivery, and utilisation of quality MNH services. Identifying the provision of health system inputs in terms of health workforce, equipment, medicine, and services is essential to track the implementation of policies and ensure the progress towards universal health coverage (UHC) and SDG3 [27]. Further analysis of nationally representative surveys (e.g., Nepal Health Facility Survey) can generate evidence on the status of health system readiness (structural quality). However, despite high policy priority on quality health care, there is limited evidence available on the status of health system readiness for MNH services and their determining factors in Nepal. Therefore, this study aimed to examine health facilities' structural quality (inputs) and their associated factors for MNH services. Findings of this study can be instrumental in planning and monitoring health facilities and provide insights to policymakers to set priorities. Furthermore, findings will help to allocate scarce resources for effective implementation of MNH policies for improved health status of mothers and newborns in Nepal.

## Methods

### Study design

This was a cross-sectional study based on further analysis of secondary data. Data for this study were derived from the nationally representative Nepal Health Facility Survey (NHFS) 2015 [28]. The detailed methodology for the NHFS 2015 has been described in its full report [28]. In NHFS 2015, health facility level information was collected using facility inventory and conducted interviews with the health facility in-charge. In addition, health workers' training and competency-related information were collected by interviews with specific health workers who provide specific health services (e.g., ANC service). For this study, data from the health facility inventory and health workers' interview files were merged using a unique health facility identifier available in each file. Health facilities and workers' information were compiled to calculate the structural quality of health facilities. The structural quality of health facilities was assessed for 901 health facilities providing ANC services, and 454 health facilities providing perinatal care services.

### Nepal's health system context for maternal and newborn health services

Nepal has three levels of government: local, provincial, and federal. Health system governance is in line with the government system. For instance, the local health system covers

community-level health facilities (e.g., community health clinics, outreach primary health and immunisation clinics) and health facilities at the ward level (e.g., health posts). In addition, primary health care centers (PHCCs) and district hospitals are also included in the local health system [24, 29]. At the community level, the network of Female Community Health Volunteers (FCHVs) supports community-based health programs, especially in providing preventive, promotive health services to women and children in their catchment areas. Community health workers (e.g., auxiliary health workers and nurse midwives) provide primary health care services in community outreach immunisation and community health clinics and health posts. Health posts offer routine MNH interventions during antenatal, facility birth, and postnatal care visits. In addition, some health posts are accredited birthing centres that provide institutional delivery services for normal pregnancies. While PHCCs and district hospitals provide basic emergency obstetric and neonatal care are the first referral health institutions. The provincial health system includes hospitals that offer tertiary services such as comprehensive emergency obstetric and neonatal care and specialist health services. Health facilities of the federal level include central level hospitals that provide tertiary and super-specialised services related to maternal and newborn health.

### Independent variables

Based on the information available in the dataset and previous studies [30–32], seven health facility level independent variables were selected, such as managing authority (Private, Public), facility types, provinces (province 1 -not named yet), Madhesh, Bagmati, Gandaki, Lumbini, Karnali, Sudurpaschim), mechanism of quality assurance (Yes, No), frequency of health facilities' management meeting (No, Sometimes, and Monthly), the existence of feedback collection system in health facilities (Yes, No), availability of external supervision of staff (Yes, No). In addition, Routine quality assurance activity was coded as "yes" for facilities reporting that it routinely carries out quality assurance activities (documentation of report or minutes of a quality assurance meeting, a supervisory checklist, a mortality review, or an audit of records or registers) and "No" for those without such quality assurance activities [28].

### Outcome variables and measurement

The antenatal period covers the time from conception to before labour pain, and perinatal services cover services provided during labor and within the first week of childbirth [33]. This study has two outcome variables: Structural quality of health facilities for i) ANC visits (poor, optimal), ii) perinatal services (poor, optimal).

In the NFHS 2015, data were collected using the World Health Organization's (WHO) Service Availability and Readiness Assessment (SARA) manual [34]. The WHO's SARA manual provides a list of items to be included in assessing the structural quality of health facilities under two domains: a) service availability and b) facility readiness [34, 35]. The service availability domain covers a list of recommended service interventions that should be available and when service users attend those health facilities (Tables A and B in S1 File). Under the domain of service availability for perinatal services, there were two subdomains: newborn care, and delivery care (Table B in S1 File). The facility readiness domain covers four sub-domains for both services: general readiness (e.g., water, electricity), equipment (e.g., delivery beds for childbirth services), medicine/commodities (e.g., misoprostol, magnesium sulphate, iron tablets), and staff and guidelines (e.g., availability of protocols, guidelines for training). Based on national guidelines for maternal and newborn care [36], and availability of information in the dataset [28], we contextualised and extracted information for the domain and subdomain-specific items for structural quality of health facility for MNH services taking reference of previous

studies [13, 28, 34, 36, 37]. Based on the information available in dataset, a number of domain and sub-domain-specific items were identified to calculate the structural quality scores of health facilities for ANC, and perinatal services (Tables A and B in S1 File). We calculated health system inputs or structural quality of health facilities considering previous studies [31, 38]. First, sub-domain-specific structural quality scores of health facilities for each service were calculated. Averaging subdomain scores, domains scores were calculated for each outcome variable (e.g., ANC services). The average scores of two domains (service availability and facility readiness) were the structural quality of health facility for MNH service. Structure and distribution (e.g., normality) of structural quality scores of health facilities were checked for regression analysis. Distribution of structural quality of health facilities for each service was skewed. Thus, we considered the mean as the cut-off point for dichotomization of score [39, 40], which allows to estimate the odds ratios (ORs) of determinants associated with structural quality of health facilities. Thus, considering the mean score as the cut-off point, the health facilities score was dichotomised into poor (if health facilities score< mean) or optimal (if health facilities score $\geq$ mean) structural quality of health facilities for each MNH service.

## Statistical analyses

Binomial logistic regression analysis was conducted to identify the health facility level determinants of the structural quality of health facilities for MNH services. Bivaraible and multivariable regression models were conducted for each outcome variable. In the descriptive analysis, frequency, mean score of structural quality of Health facilities for both services, proportion, p values obtained from the chi-square association of each independent variable and outcome variable were reported. The statistical significance level was p<0.05 (two-tailed). Before running the multivariable regression (back ward elimination) model, multicollinearity was checked and excluded independent variables having variation inflation factor $\geq$3 in the multivariable regression analyses [41]. The model fitness test was conducted using the Hosmer Lemeshow test (non-significant results (p>0.05) indicated an adequate fit) [42]. All estimates were weighted otherwise indicated. In addition, we adjusted the clustering effects of sampling design in the data analysis stage using the clients' weight and accounting for survey strata: region and types of health facilities. All analyses were conducted using the survey (svy) command function and considering the clustering effect in Stata 14.0 (Stata Corp, 2015).

## Ethics approval

We used secondary data from the 2015 NHFS. This survey was approved by an ethical review board of Nepal Health Research Council, Nepal, and ICF Marco International, Maryland, USA. The Ministry of Health and Population (MOHP) (Nepal) oversaw the overall research process of the NHFS 2015. The NHFS data are publicly available for further analysis, and data were deidentified of the research participants. This study did not require ethical approval from respective institutions. However, the first author took approval for the download and use of the dataset for his doctoral thesis and this publication.

## Results

### Descriptive analysis of health facilities providing routine MNH services

Table 1 shows the descriptive characteristics of health facilities providing ANC, and perinatal services. Of 901 health facilities providing ANC services, more than nine in ten (93%) were managed by the public sector. Nearly nine in ten (86%) were peripheral level health facilities (health posts and clinics). More than half (52%) of the health facilities were in the Hill region.

**Table 1. Descriptive characteristics of health facilities with MNH services in Nepal, 2015.**

| Determinants | Categories | Facilities providing ANC services (N = 901) | | Facilities providing perinatal services (N = 454) | |
|---|---|---|---|---|---|
| | | Frequency | % | Frequency | % |
| Managed by | Private | 64 | 7.1 | 45 | 9.9 |
| | Public | 837 | 92.9 | 409 | 90.1 |
| Facility types | PHCCs and hospitals | 122 | 13.5 | 105 | 23.1 |
| | Health posts and clinics | 779 | 86.5 | 349 | 76.9 |
| Region | Mountain | 112 | 12.4 | 66 | 14.6 |
| | Hill | 473 | 52.5 | 275 | 60.6 |
| | Terai | 316 | 35.1 | 112 | 24.7 |
| Province | One | 160 | 17.8 | 78 | 17.1 |
| | Madhesh | 154 | 17.1 | 38 | 8.5 |
| | Bagmati | 179 | 19.8 | 80 | 17.7 |
| | Gandaki | 116 | 12.9 | 66 | 14.5 |
| | Lumbini | 135 | 15.0 | 64 | 14.0 |
| | Karnali | 68 | 7.5 | 60 | 13.3 |
| | Sudurpaschim | 88 | 9.8 | 67 | 14.9 |
| Health facility meeting | No | 165 | 18.3 | 75 | 16.6 |
| | Sometimes | 129 | 14.4 | 71 | 15.6 |
| | Monthly | 607 | 67.4 | 308 | 67.8 |
| Quality assurance activities | No | 714 | 79.3 | 360 | 79.4 |
| | Yes | 187 | 20.7 | 93 | 20.6 |
| Feedback collection | No | 489 | 54.2 | 222 | 48.9 |
| | Yes | 412 | 45.8 | 232 | 51.1 |
| Supervision of staff | No | 330 | 36.6 | 128 | 28.2 |
| | Yes | 571 | 63.4 | 326 | 71.8 |

ANC: Antenatal care; PHCCs: Primary health care centers

Nearly four in five (79%) health facilities did not have quality assurance activities or feedback collection systems within the past year. However, two-thirds (67%) had monthly facility management meetings and external supervision visits in the past four months.

Of the 454 health facilities assessed for perinatal services, nine in ten (90%) health facilities were peripheral health facilities (health posts and health clinics). Public authorities managed more than three in four (90.1%) health facilities. However, nearly eight in ten (79%) did not have a quality assurance system in the past year. In contrast, two-thirds of health facilities had external supervision and had a facility management meeting (Table 1).

## Services availability and facility readiness items for MNH services

Table 2 shows the service availability and readiness items of health facilities (N = 901) for ANC service in Nepal. Of items included in the availability of the services, there were low items available in laboratory-related items such as tests for urine test (14.2%), blood for haemoglobin (8.1%), and anaemia (17.9%). In the subdomains of facility readiness domain, health facilities were poorly equipped with staff and guidelines, including ANC screening training (13.8%). In addition, there was low availability of medicine such as misoprostol tablets (17.1%), and equipment such as digital blood pressure tool (2.2%). Only 12.3% of health facilities had 24-hour staff availability for ANC services.

**Table 2. Service availability and facility readiness assessment items in health facilities for ANC services (N = 901).**

| Service availability domain: | | |
|---|---|---|
| **Services availability items** | **Frequency** | **Yes (%)** |
| ANC counselling | 898 | 99.7 |
| Birth preparedness package counselling | 890 | 98.8 |
| Albendazole tablets distribution | 883 | 98.0 |
| Newborn care counselling | 869 | 96.4 |
| Family Planning counselling | 866 | 96.1 |
| Breastfeeding counselling | 866 | 96.1 |
| PNC counselling | 862 | 95.7 |
| Tetanus toxoid service | 835 | 92.7 |
| Blood pressure measure service | 809 | 89.8 |
| Weighting clients | 799 | 88.7 |
| HIV prevention counselling | 797 | 88.5 |
| Iron tablet distribution | 628 | 69.8 |
| Folic acid distribution | 541 | 60.0 |
| HIV test and counselling | 319 | 35.5 |
| Measure height | 207 | 22.9 |
| Health education service | 164 | 18.2 |
| Misoprostol distribution | 154 | 17.1 |
| Anaemia test service | 161 | 17.9 |
| Urine protein test | 128 | 14.2 |
| Urine test service | 91 | 10.1 |
| Haemoglobin test services | 73 | 8.1 |
| **Facility readiness domain** | | |
| **General readiness** | | |
| Client latrine | 736 | 81.7 |
| Client waiting area | 719 | 79.8 |
| Water supply | 730 | 81.0 |
| Electricity service | 573 | 63.6 |
| Emergency transport | 536 | 59.5 |
| Landline phone | 140 | 15.6 |
| 24-hour staff availability | 111 | 12.3 |
| **Medicine** | | |
| Albendazole tablets | 883 | 98.0 |
| Tetanus toxoid vaccine | 835 | 92.7 |
| Iron-folic tabs | 628 | 69.8 |
| Folic acid | 541 | 60.0 |
| Misoprostol tablets | 154 | 17.1 |
| **Equipment** | | |
| Examination table | 838 | 93.1 |
| Autoclave service | 836 | 92.8 |
| Fetoscope available | 835 | 92.7 |
| Weighing scale | 817 | 90.7 |
| Stethoscope | 813 | 90.3 |
| Blood pressure set manual | 797 | 88.5 |
| Thermometer | 739 | 82.0 |
| Disinfectant for Infection Prevention | 598 | 66.4 |
| Soap for Infection Prevention | 496 | 55.0 |

(*Continued*)

**Table 2.** (Continued)

| Water for infection prevention | 445 | 49.4 |
|---|---|---|
| Examination light | 424 | 47.1 |
| Tape fundal height | 272 | 30.2 |
| Digital blood pressure tool | 20 | 2.2 |
| **Staff training and guidelines** | | |
| Supervision of staff | 751 | 83.3 |
| IEC materials for ANC service | 621 | 68.9 |
| ANC guideline | 217 | 24.0 |
| Complication and management | 134 | 14.9 |
| ANC counselling training | 131 | 14.5 |
| ANC screening training | 125 | 13.8 |
| Nutritional assessment | 82 | 9.1 |
| Other training (e.g., refresher training on ANC) | 21 | 2.3 |

ANC: Antenatal Care; PNC: Postnatal Care, IEC: Information, and Communication

Similarly, Table 3 shows the subdomain-specific items available in health facilities (N = 454) for perinatal services. Under the service availability domain, two in five health facilities had availability of injectable medicine for mothers (41%) and parental convalescent such as magnesium sulphate (9.6%). Nearly one in four (23.7%) health facilities had 24-hour staff for perinatal services and poor availability of communication services including landline services (23.2%) and mobile services (11.5%). In addition, health facilities had poorly equipped with trained staff and guidelines such as neonatal sepsis management (19.4%). Furthermore, health facilities had a low stock of equipment and commodities such as alcohol for hand rubs (26.1%) and availability of essential medicine such as Nifedipine capsule (19.1%) and calcium gluconate (26.4%).

## Structural quality of health facilities for MNH services

Fig 1 shows the structural quality scores of health facilities for the first ANC visit. The mean score of the structural quality of health facilities for ANC services was 0.62 (maximum: 1.00), with service availability scoring 0.67 and facility readiness scoring 0.56. Out of four subdomains of facility readiness, the staff and guidelines subdomain had the lowest score (0.31), whereas the highest score (0.72) was for equipment.

Similarly, Fig 2 shows the structural quality score of health facilities for perinatal services. The average structural quality score of health facilities with perinatal services was 0.67 (maximum: 1.00), with higher scores for service availability (0.72) than facility readiness (0.62). The staff and guidelines' mean score for the facility readiness subdomain was lower (0.31) than equipment (0.82) for perinatal services.

## Distribution of health facilities with optimal structural quality for MNH services

Table 4 shows the distribution of the structural quality of health facilities for MNH services in Nepal. Higher-level health facilities (PHCCs and hospitals) had the highest percentage of the optimal structural quality of health facilities for ANC services (64%) compared to peripheral health facilities (e.g., health posts and clinics) (18%). Staff supervised in the past four months (29%) demonstrated optimal structural quality of health facilities for ANC services than staff

**Table 3. Service availability and facility readiness assessment items of health facilities for perinatal services (N = 454).**

| Service availability domain | | |
|---|---|---|
| **Newborn care services** | **Frequency** | **Yes (%)** |
| Immediate breastfeeding | 450 | 99.1 |
| Wrapping baby | 443 | 97.6 |
| Weighing newborn | 434 | 95.8 |
| Head to toe examination | 429 | 94.6 |
| Kangaroo mother care | 415 | 91.5 |
| Skin to skin contact | 413 | 91.1 |
| Delayed bathing | 308 | 67.9 |
| Use of chlorhexidine | 289 | 63.6 |
| Newborn resuscitation | 167 | 36.9 |
| Injectable antibiotic available | 186 | 41.0 |
| **Maternity care services** | | |
| Oxytocin parental | 390 | 86.0 |
| Use of paratograph | 384 | 84.7 |
| Injectable antibiotic available | 186 | 41.0 |
| Antibiotics parental | 184 | 40.7 |
| Anticonvulsant parental | 44 | 9.6 |
| **Facility readiness domain** | | |
| **General readiness** | | |
| Client latrine | 411 | 90.6 |
| Protected client waiting area | 393 | 86.6 |
| Water supply | 386 | 85.0 |
| Electricity service | 295 | 65.1 |
| Emergency transport | 282 | 62.2 |
| 24-hour duty call | 107 | 23.7 |
| Landline phone | 105 | 23.2 |
| Mobile phone | 52 | 11.5 |
| **Medicines** | | |
| Betadine solution | 415 | 91.6 |
| Intravenous fluid | 410 | 90.5 |
| Tablet oxytocin | 402 | 88.5 |
| Tablet magnesium sulphate | 329 | 72.6 |
| Chlorhexidine tube | 263 | 58.0 |
| Injectable antibiotics | 186 | 41.0 |
| Calcium gluconate | 120 | 26.4 |
| Nifedipine capsule | 87 | 19.1 |
| **Equipment** | | |
| Autoclave services | 450 | 99.2 |
| Delivery bed | 440 | 96.9 |
| Fetescope | 419 | 92.4 |
| Latex gloves | 421 | 92.8 |
| Infant scale | 414 | 91.3 |
| Sponge holder | 416 | 91.8 |
| Stethoscope | 405 | 89.2 |
| Delivery set | 415 | 91.6 |
| Cord cutting blade | 411 | 90.7 |

*(Continued)*

**Table 3.** (Continued)

| | | |
|---|---|---|
| Needle holder | 400 | 88.1 |
| Suturing blade | 386 | 85.0 |
| Bag and Mask | 380 | 83.8 |
| Epitomy set | 376 | 82.8 |
| Blood pressure set | 380 | 83.9 |
| Forceps | 378 | 83.4 |
| Disinfectant | 364 | 80.3 |
| Blank paratograph | 363 | 80.1 |
| Baby wrappers four sets | 306 | 67.6 |
| Thermometer | 361 | 79.6 |
| Vaginal speculum | 354 | 78.1 |
| Soap available in the maternity room | 326 | 72.0 |
| Cord clamper | 323 | 71.2 |
| Water available in the delivery room | 313 | 69.1 |
| Nayano Jhola set | 308 | 68.0 |
| Examination light | 293 | 64.7 |
| Dee-lee suction | 234 | 51.7 |
| Alcohol for hand rub | 118 | 26.1 |
| **Staff training and guidelines** | | |
| Supervision of health workers | 409 | 90.1 |
| External supervision in the last four months | 326 | 71.8 |
| Exclusive breastfeeding training | 142 | 31.3 |
| Neonatal resuscitation training | 136 | 29.9 |
| Kangaroo Mother Care training | 128 | 28.2 |
| Reproductive health guideline | 128 | 28.2 |
| Cord cutting training | 127 | 27.9 |
| Integrated management of pregnancy and childbirth | 123 | 27.2 |
| Acute management of the third stage of labour | 124 | 27.4 |
| Thermal care | 120 | 26.4 |
| Routine labour and delivery | 118 | 25.9 |
| Maternal and newborn care update emergency obstetric care | 97 | 21.4 |
| Neonatal sepsis management | 88 | 19.4 |
| Other training (e.g., refresher training) | 6 | 1.4 |

without such supervision (16%). Private health facilities (57%) had optimal structural quality of health facilities for ANC services compared to public facilities (22%) (Table 4).

Private health facilities (49%) had optimal structural quality for perinatal services than public facilities (22%). On the other hand, the health facilities of the Karnali province (12%) had poor structural quality for perinatal services compared to province one (38%).

## Determinants of the optimal structural quality of health facilities for MNH services

In the bivariable analysis, out of the seven independent determinants examined, five determinants, including structural (management authority), intermediary (types of health facilities, province), and health system (health facility meeting, supervision of staff in the past four months), were significantly associated with optimal structural quality of health facilities for ANC services. However, in the multivariable analysis, private health facilities had higher odds

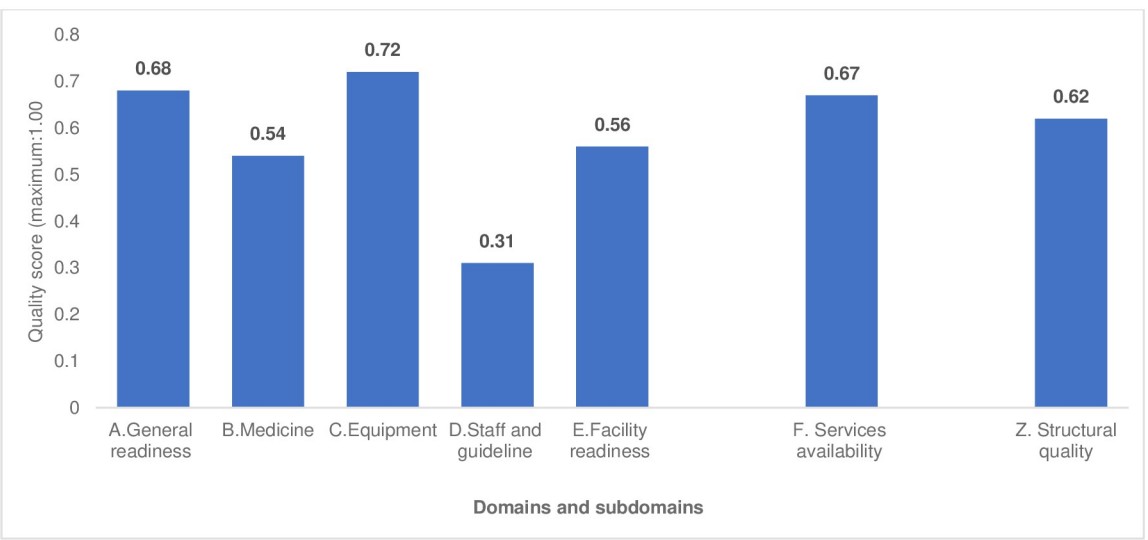

**Fig 1. Structural quality score of health facilities for the ANC visit.**

of (aOR = 2.65, 95% CI: 1.48, 4.74) optimal structural quality than public facilities. Health facilities with external supervision in the past four months were more likely (aOR = 1.96, 95% CI: 1.22, 3.13) to have optimal structural quality for ANC services than health facilities without supervision. The peripheral health facilities (e.g., health posts) had poor structural quality for ANC services; for instance, the odds of optimal structural quality were 87% lower in peripheral health facilities compared to higher-level health facilities.

In the bivariable analysis, out of the seven independent determinants, three intermediary (types of health facilities, province, and region) and one health system (feedback collection) determinants were significantly associated with optimal structural quality of health facilities

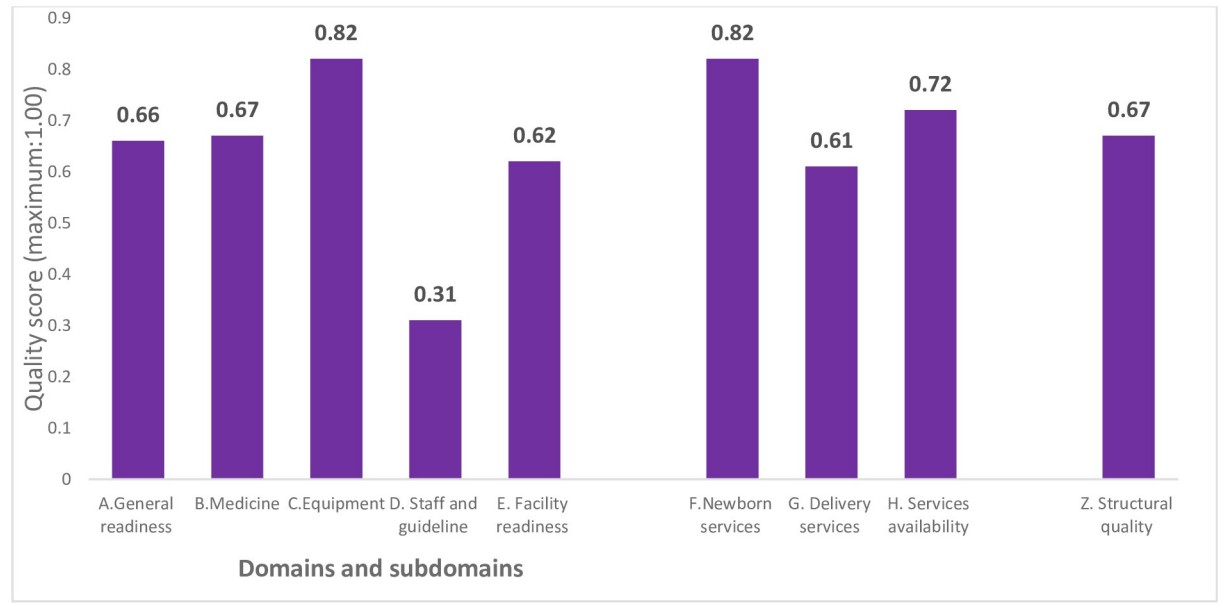

**Fig 2. Structural quality score of health facilities for the perinatal services.**

**Table 4. Distribution of optimal structural quality of health facilities for routine MNH services in Nepal, 2015.**

| Determinants | Categories | Health facilities providing ANC services (N = 901) | | | Health facilities providing perinatal services (N = 454) | | |
|---|---|---|---|---|---|---|---|
| | | Frequency | Optimal quality (%) | p | Frequency | Optimal quality (%) | P |
| National average | | 901 | 24.4 | | 454 | 29.3 | |
| Managed by | Private | 64 | 56.7 | <0.001 | 45 | 49.5 | <0.001 |
| | Public | 837 | 22.0 | | 409 | 27.0 | |
| Facility type | PHCCs and hospitals | 122 | 64.1 | <0.001 | 105 | 58.6 | <0.001 |
| | Health posts and clinics | 779 | 18.2 | | 349 | 20.4 | |
| Provinces | One | 160 | 19.8 | 0.192 | 78 | 38.3 | <0.001 |
| | Madhesh | 154 | 18.2 | | 38 | 48.7 | |
| | Bagmati | 179 | 31.4 | | 80 | 26.2 | |
| | Gandaki | 116 | 22.7 | | 66 | 18.0 | |
| | Lumbini | 135 | 28.9 | | 64 | 42.6 | |
| | Karnali | 68 | 20.1 | | 60 | 12.6 | |
| | Sudurpaschim | 88 | 28.3 | | 67 | 24.7 | |
| Quality assurance activities | No | 714 | 23.1 | 0.138 | 360 | 27.6 | 0.161 |
| | Yes | 187 | 29.5 | | 93 | 35.6 | |
| health facility meeting | Never | 165 | 16.3 | 0.076 | 75 | 22.8 | 0.479 |
| | Sometimes | 129 | 24.7 | | 71 | 29.1 | |
| | Monthly | 607 | 26.6 | | 308 | 30.9 | |
| Feedback collection | No | 489 | 21.2 | | 222 | 21.1 | |
| | Yes | 412 | 28.3 | 0.052 | 232 | 37.0 | 0.002 |
| Supervision of staff | No | 330 | 16.4 | <0.001 | 128 | 24.1 | 0.199 |
| | Yes | 571 | 29.1 | | 326 | 31.3 | |

PHCCs: Primary Health Care Centers; ANC: Antenatal Care

for perinatal services. However, in the multivariable analysis, two determinants (types of health facilities, and province) were significantly associated with optimal structural quality of health facilities for perinatal services. The odds of optimal structural quality of health facilities for perinatal services were 84% lower (aOR = 0.16; 95% CI: 0.10, 0.27) in peripheral health facilities, compared with higher-level health facilities. Similarly, health facilities of Karnali province had 71% lower odds of having optimal structural quality of health facilities for perinatal services compared to province one (Table 5).

## Discussion

This study used facility inventory and health workers interview data from the Nepal Health Facility Survey 2015 examined the health facility readiness for MNH services in Nepal. Overall, the availability of trained workforces, and laboratory-related facilities was low than other sub-domains of SARA framework. In addition, this study revealed suboptimal structural quality of health facilities for MNH services. While health facilities supervised by a higher authority had optimal quality ANC services, peripheral health facilities had poor quality ANC services. On the other hand, health system readiness in private health facilities had optimal quality for ANC, and perinatal care services. In addition, Karnali province and peripheral areas' public and private health facilities had poor quality perinatal services.

Health facilities had poor availability of trained workforce, laboratory, and general readiness (e.g., mobile communication, ambulance) for ANC and perinatal services. Findings of

**Table 5. Determinants associated with optimal structural quality of health facilities for routine MNH services in Nepal, 2015.**

| Determinants | Categories | Health facilities providing ANC services (N = 901) | | Health facilities providing perinatal services (N = 454) | |
|---|---|---|---|---|---|
| | | cOR (95% CI) | aOR (95% CI) | cOR (95% CI) | aOR (95% CI) |
| Managing authority | Public | 1.00 | 1.00 | 1.00 | 1.00 |
| | Private | 4.64(2.88, 7.50) *** | 2.65(1.48, 4.74) ** | 2.60 (1.35, 4.53) *** | 1.69 (1.25, 3.40) ** |
| Facility type | PHCCs and hospitals | 1.00 | 1.00 | 1.00 | 1.00 |
| | Health posts and clinics | 0.12(0.09, 0.18) *** | 0.13(0.09, 0.18) *** | 0.18(0.12, 0.28) *** | 0.16 (0.10, 0.27) *** |
| Province | One | | | 1.00 | 1.00 |
| | Madhesh | 0.90 (0.43, 1.89) | | 1.53 (0.60, 3.88) | 0.86(0.24, 3.11) |
| | Bagmati | 1.86(1.02, 3.36) * | | 0.57(0.28, 1.17) | 0.47(0.20, 1.07) |
| | Gandaki | 1.19(0.60, 2.35) | | 0.35(0.13, 0.96) * | 0.39(0.12, 1.24) |
| | Lumbini | 1.65(0.88, 3.09) | | 1.19(0.55, 2.60) | 1.20(0.47, 3.06) |
| | Karnali | 1.02(0.45, 2.29) | | 0.23(0.08, 0.66) ** | 0.29(0.09, 0.99) * |
| | Sudurpaschim | 1.60(0.80, 3.17) | | 0.53(0.24, 1.16) | 0.53(0.22, 1.29) |
| Quality assurance activities | No | 1.00 | | 1.00 | |
| | Yes | 1.39(0.90, 2.17) | | 1.45(0.86, 2.43) | |
| health facility meeting | No | 1.00 | | 1.00 | |
| | Sometimes | 1.68(0.85, 3.35) | | 1.39(0.59, 3.25) | |
| | Regularly | 1.86(1.07, 3.24) * | | 1.52(0.77, 2.99) | |
| Feedback collection | No | 0.68(0.46, 1.00) | | 0.46(0.28, 0.75) ** | |
| Supervision | No | 1.00 | 1.00 | 1.00 | |
| | Yes | 2.10(1.37, 3.22) *** | 1.96(1.22, 3.13) ** | 1.44(0.82, 2.50) | |

Significance at

*** p<0.001,

** p<0.01,

* p<0.05.

The goodness of fit test (Hosmer Lemeshow test, p = 0.766) for the regression model of outcome structural quality for ANC service. The goodness of fit test (Hosmer Lemeshow test, p = 0.199) for the second outcome variable, i.e., structural quality for perinatal services.

PHCCs: Primary Health Care Centers; ANC: Antenatal Care.

poor general readiness were consistent with the previous study conducted in Karnali province, which showed shortage of medicine for women and newborns [43] and many low and low income countries of South Asia and Africa [44]. Competent staff and equipment availability are important components of health facility readiness for quality service delivery [45]. For instance, attending health facilities for ANC is to screen pregnant women with possible complications and timely referral, but without trained workforces and needed equipment to screen difficult to screen complicated pregnancies [45–48]. These are crucial to identifying and managing potential pregnancy and childbirth complications. In addition, availability of trained workforces, equipment, and laboratory services helps build trust with the health system and increase the service users' engagement with the system [43, 49]. Thus, local health system authorities need to identify competent staff and equipment availability in their catchment health facilities and ensure optimal health system readiness.

Of the domains listed in the SARA manual, the lowest scores were observed for the staff training and guidelines subdomain in all health facilities for ANC, and perinatal services. A study conducted in Southern Nepal also revealed that less than half of the health workers had received the mandated skilled birth attendants training [45]. Reasons behind low scores on this subdomain could be less focused on compliance with standard protocols and continuing

education. Health workers focus on training, but low implementation of skills gained after attending training. In addition, there is social desirability of getting more training in some cases if they reported they had not received training in the interview response. The health workforce is vital for optimal health system readiness and quality MNH service delivery. Possible strategies for optimal health facility quality in the staff and guideline subdomain, could be improving the skills of the health workforce through training on essential MNH services and providing materials and guidelines for specific health services [14, 50]. Moreover, optimal readiness for the staff and guideline subdomain can be strengthened by ensuring the supply of essential medicines and equipment at the health facilities. In Nepal, the federal health system provides an opportunity and the resources to strengthen the inputs, such as recruitment of health trained health workforces and supply of training materials at health facilities through effective collaboration with provincial and municipal governments at the local level [51].

In this study, private health facilities had two-fold higher odds of having optimal structural quality for ANC services, and perinatal services compared to public health facilities. Previous studies showed high-quality scores for primary health care services compared to public facilities in Nepal [31] and Bangladesh [52]. Private health facilities are usually urban-centric, and have health infrastructure, equipment and supplies, and availability of health workforce [53]. In private health facilities, compared to public facilities, the client flow is generally low [54], are more responsive, hospitable and client-oriented [55], and have short waiting time [56]. On the other hand, public health facilities are often compromised by inadequate inputs, including human workforce, equipment, and medicine, with adverse effects on health facilities' readiness [57]. While private health facilities offer optimal quality ANC services, users also incur high out-of-pocket expenses, including routine MNH services, free at the point of services covered by government funding. In urban or remote areas, women with lower socioeconomic status have limited access to private health services in Nepal, increasing client flow to public facilities, which results in overcrowding and receiving poor quality of care [58]. According to private hospitals' operational guidelines, private health facilities should allocate 10% of beds to disadvantaged populations [24]. Although there are increasing trends in the utilisation of maternal and child health services in Nepal over the last two decades [55], there are still no functional monitoring mechanisms to evaluate if this is implemented [58, 59]. Proper monitoring and facilitation of the implementation of this policy provision could increase access to private health facilities, especially for women of marginalised groups in urban areas. Private health facilities are also eligible to participate in the Government's Safe Delivery Incentive Program. This maternity incentive program provides a monetary incentive to women who complete 4ANC visits or give birth at health facilities [60], reimburses the health service provider for services delivered, and provides health facilities with a financial incentive in cases of cesarean delivery. Very few private health facilities participate in this program; however, the amount reimbursed is lower than the private health facility charge. Private health facilities also have high rates of caesarian section delivery [61]. High care costs of routine health services in private health facilities partly contribute to Nepal's high OOP expenditure. Thus, access to private maternity services could be improved through the linkage of the national health insurance program with the private health providers [62], where women can get maternity services, and health insurance program can reimburse the cost of health care in private health facilities. Nonetheless, more than two thirds of women received maternity services from public faculties in Nepal, mostly by women with lower socioeconomic status, and ethic disadvantaged women [60]. Therefore, improving quality in public facilities is vital to reduce the maternal and neonatal deaths.

The study showed peripheral health facilities had poor structural quality for ANC, and perinatal care services. In contrast, health facilities of Karnali province had poor structural quality

perinatal services, which are likely to result in poor quality MNH services. These findings resonate with available evidence; for instance, past studies suggest that peripheral health facilities were poorly prepared for quality primary health services in Nepal [30–32], India [63], and Burkina Faso [64]. This is of concern as peripheral health facilities provide most routine ANC, childbirth PNC services [60]. During pandemic, the flow of health services users decreased in referral health facilities such as tertiary referral hospitals, and services in rural/peripheral facilities increased. The findings of this study highlight the need for improvement in the quality of care in peripheral facilities for better MNH outcomes. For decades, the Karnali province has had difficult geographical settings, poor roadworks, and neglected mainstream development. Evidence shows that rural areas and Karnali province had a high burden of maternal and neonatal morbidity and mortality [6] attributed in part to poor quality of MNH care [65], due to a lack of skilled birth attendants in health facilities [47, 66], and inadequate skills to handle MNH complications at peripheral level [49]. Previous studies have also revealed poor MNH outcomes in peripheral and remote areas in Karnali province, in part due to suboptimal health system readiness [30], poor access to quality care [66, 67], and lack of transportation and health facility accessibility [12]. Improving health system readiness needs tailored and customised approaches according to the context. Contextual approaches can be implemented, including recruitment of local health workers and support from local governments in infrastructure development, supplies of medicines and equipment. Thus, policy and program initiatives focus on improving the health facility capacity of remote regions and peripheral health facilities to deliver optimal quality MNH services.

Health facilities supervised by higher authorities had better health system readiness for ANC services. Supervision and monitoring of health facilities in peripheral areas could improve the quality of MNH services in two ways [58]: improving the technical quality of health services and health management functions. Technical mentors could transfer technical skills, observe procedures during monitoring and supervisory visits and provide inputs to improve health workers' skills. Available evidence from Nepal shows onsite coaching and mentoring from a higher-level authority can improve the quality of MNH services [68]. Management mentors could identify enabling environment of health facilities, including community support, internal inputs, management function such as monthly meetings, and quality assurance mechanisms in the health facilities. However, Nepal's difficult geographical terrain might be unfeasible for in-person training at the district level. In this context, periodic supervision visits, onsite coaching, and mentoring support to peripheral health facilities and health workers could be important strategies for better health system readiness for quality MNH service delivery.

## Implications for policy and programmes

This study has implications for programs and policies. First, this study identified the poor structural quality of health facilities in Karnali province and peripheral health facilities and public health facilities. Provincial and local municipal governments should prioritise health system inputs, including local health workforce recruitment, and supplies of necessary health commodities and medicines. In Nepal's federal health system, local governments (municipalities) have autonomy and budgets to address contextual problems, recruit health workforces, and improve essential medicines and supplies. Second, local rural municipalities should supervise and monitor peripheral health facilities. The health section of municipalities could monitor and supervise ward-level health facilities such as health posts, community health clinics in their catchment. Monitoring and supervision visits from the local health officer of municipality could improve the local health workers' health management functions and technical skills. Local health offices can use service availability and facility readiness framework and identify

the availability of subdomain-specific items during the monitoring and supervision of the health facilities. A study from Pakistan revealed supportive supervision, recognition, training, logistics, and salaries were community and health system motivating factors for lady health supervisors, and motivated by their role in providing supportive supervision and supervisory support from their coordinators and managers [69].

Third, private health facilities have optimal structural quality for ANC services, but disadvantaged women have poor access to private health services. Implementing maternity incentive programs in private health facilities, cost-sharing and ensuring allocation of 10% beds in private health facilities for disadvantaged populations could increase the access to private maternity services, especially for disadvantaged women in urban areas. Fourth, this study also highlighted using the SARA manual to collect input information during routine supervision and monitoring visits from higher-level health facilities. Later such information can be used to calculate the subdomain-specific health system response for quality MNH services. Finally, this study used multiple data sources and calculated the quality score at the health facilities level covering multiple dimensions. There are data available at the local level, and local health facility managers can also use data from multiple sources to identify the quality index of health facilities in their catchment.

## Strengths and limitations

This study has some strengths. First, this study analysed the nationally representative survey data and assessed the availability and structural quality of health facilities for MNH services in Nepal. So, the findings of this study are generalisable for all regions of Nepal. Second, this important study considered a wide range of health items needed to deliver routine MNH services. The SARA manual, other guidelines on maternity care, and national standard recommended several items/interventions needed to deliver quality health service. This study accounted wide range of items to estimate the quality score. Third, we used multiple data sources, and identified the composite quality scores of health facilities based on the data derived from observation, such as medicines and equipment in health facilities by a trained enumerator. Therefore, findings might be more reliable compared to the perceived quality of care assessment. Limitations of this included, first, we analysed data from NHFS 2015 conducted five years earlier; therefore, the data may not reflect recent conditions of Health facilities in the federal health system context of Nepal. However, this study used recent nationwide health facility survey data, thus can give the overall picture of health system readiness for MNH services. Second, NHFS is a cross-sectional survey; the inferences indicate the correlation rather than causality. Third, the outcome variable's score distribution did not allow us to run the linear regression. Therefore, due to data distribution and structure, we dichotomised scores taking the mean as cut-off point to run logistic regression [39]. Finally, measuring facility readiness and structural quality is difficult. Some researchers have raised concerns about which items are included (vs. excluded) in the creation of scores [70, 71], as well as concerns about the poor correlation of readiness scores with observed service quality [72]. However, adapting the SARA framework [34], national standards [36], and previous studies [13, 38, 73], we created score of structural quality of health facilities for MNH services. Using secondary data for the analysis always has its limitations, and important information might not be available for analysis. Nevertheless, we have included all available information and analysed the recent national-level facility survey data. Findings and methods used in this analysis could be a reference for future research. Authors' experience with the health system also suggests that Nepalese health facilities are constrained from many health systems inputs, shortage of medicine, equipment, health workforce and general readiness. Finally, this quantitative study could

not provide underling factors of suboptimal quality of care, thus, future qualitative studies can explore contextual factors associated with maternal continuum of care.

## Conclusions

Health facilities in Nepal had sub-optimal structural quality of MNH services across the continuum of care, especially health facilities in rural areas and publicly managed. Health facilities were poorly equipped with staff, training, and laboratory-related equipment and services. Private health facilities and health facilities supervised by higher authorities had optimal structural quality for MNH services, while peripheral health facilities, and health facilities of Karnali province had poor structural quality for MNH services. Maternity and newborn incentive programs such as maternity and newborn incentive program can be implemented in private health facilities to use maternity services in those facilities at the subsidised cost reimbursed by the program. There is an urgent need for policy reform to improve the MNH services, particularly in the public and health facilities of Karnali province. Provincial and local governments should focus on improving the health system inputs, including trained health workers, supply of essential medicines, and provision of laboratory-related equipment in those areas. Achieving universal health coverage will not be possible without strengthening service availability and facility readiness in public and peripheral health facilities.

## Supporting information

**S1 File. Services availability and facility readiness items for MNH services in Nepal.**
Table A: Service availability and readiness assessment items for ANC services. Table B: Service availability and readiness assessment items for perinatal services.
(DOC)

## Acknowledgments

This manuscript is a part of the first author's (RBK) doctoral thesis at the School of Public Health, the University of Queensland. To undertake the PhD degree, RBK was supported by the Research Training Program, UQ International, and Career Development Extension Scholarships funded by the Commonwealth Government of Australia and the University of Queensland, Brisbane, QLD, Australia. We gratefully acknowledge the commitment of the Australian Government and the University of Queensland, Brisbane, QLD, Australia, to their research efforts. We thank the Demographic and Health Survey program for providing access to the data sets.

## Author Contributions

**Conceptualization:** Resham B. Khatri, Yibeltal Assefa, Jo Durham.

**Data curation:** Resham B. Khatri.

**Formal analysis:** Resham B. Khatri.

**Investigation:** Resham B. Khatri.

**Methodology:** Resham B. Khatri, Yibeltal Assefa, Jo Durham.

**Project administration:** Resham B. Khatri.

**Resources:** Resham B. Khatri.

**Software:** Resham B. Khatri.

**Supervision:** Yibeltal Assefa, Jo Durham.

**Validation:** Resham B. Khatri, Yibeltal Assefa, Jo Durham.

**Visualization:** Resham B. Khatri.

**Writing – original draft:** Resham B. Khatri.

**Writing – review & editing:** Resham B. Khatri, Yibeltal Assefa, Jo Durham.

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
