## [Decision Letter · Decision Letter 0]

22 Aug 2022

PGPH-D-22-00510

Health system readiness (structural quality) of HFs for routine maternal and newborn health services in Nepal: Evidence from nationally representative health facility survey

Dear Dr. Khatri

Thank you for submitting your manuscript to PLOS Global Public Health. After careful consideration, we feel that it has merit but does not fully meet PLOS Global Public Health’s publication criteria as it currently stands. Therefore, we invite you to submit a revised version of the manuscript that addresses the points raised during the review process.

We look forward to receiving your revised manuscript.

Kind regards,

Augustine D. Asante

Academic Editor

Journal Requirements:

1. Please provide separate figure files in .tif or .eps format and remove the embedded in the manuscript file.

2. We have noticed that you have uploaded Supporting Information files, but you have not included a list of legends. Please add a full list of legends for your Supporting Information files after the references list. 

Additional Editor Comments (if provided):

Reviewers' comments:

Reviewer's Responses to Questions

**Comments to the Author**

1. Does this manuscript meet PLOS Global Public Health’s publication criteria? Is the manuscript technically sound, and do the data support the conclusions? The manuscript must describe methodologically and ethically rigorous research with conclusions that are appropriately drawn based on the data presented.

Reviewer #1: Yes

Reviewer #2: Yes

Reviewer #3: Yes

Reviewer #4: Yes

Reviewer #5: Yes

2. Has the statistical analysis been performed appropriately and rigorously?

Reviewer #1: Yes

Reviewer #2: Yes

Reviewer #3: Yes

Reviewer #4: I don't know

Reviewer #5: Yes

3. Have the authors made all data underlying the findings in their manuscript fully available (please refer to the Data Availability Statement at the start of the manuscript PDF file)?

Reviewer #1: Yes

Reviewer #2: Yes

Reviewer #3: Yes

Reviewer #4: Yes

Reviewer #5: Yes

4. Is the manuscript presented in an intelligible fashion and written in standard English?

Reviewer #1: Yes

Reviewer #2: Yes

Reviewer #3: Yes

Reviewer #4: Yes

Reviewer #5: Yes

5. Review Comments to the Author

Reviewer #1: Your paper presents an important work.

General comment: You cannot incorporate a new finding in the discussion part that has never existed in the result section. Discussion section should be a condensed & summarized form of the results section, NOT vice versa! Again better you strengthen your discussion by discussing with similar studies conducted elsewhere and justify some inconsistencies in your study as compared with others.

Line number 47: Change to: ….were assessed for structural quality of ANC …

Line number 69-70: I didn’t see training is a problem in the result section above.

Line 102: Check the bracket for references

Line 136: citation

Line 139: ….on the status …

Line 160: change was to were

Line 194-195: How did you dichotomize (Poor Better)? Do you have any reference? If so, would you cite it?

Line 194-195: why did you select ANC first as quality indicator? Why not ANC 4? Do you have a reason?

Line 258 :( 76/9 %????) Make it clear

Line 271-272: what it mean by low availability (write the actual figure.)

Table 2: Why you over looked the general readiness assessment result. eg. Mobile phone

Again in the era of Covid 19, how you overlooked about the availability of alcohol for hand rub?

Table 4: Check N for provinces

Check for the order of the tables

On discussion part, in the first paragraph the last two sentences look recommendations. Why?

Again why you overlooked the general assessment result both in discussion and conclusion section?

Reviewer #2: This manuscript focuses on an important subject regarding the availability and quality of service provided to women and newborns in health facilities in Nepal providing antenatal care, delivery and postnatal care. The paper is technically sound and coherent, and the results support the conclusion.

However, the source of data is a national survey conducted in 2015. While the use of secondary data for research can be very useful in providing good insights as they often have a wide sample size that is representative of the country in focus, but they often report data from a wide time period (usually 4-5 years). In addition, the data source can sometimes be outdated if a newer survey has been conducted. I conducted an online review and observed that there is a more recent preliminary survey report Nepal Health Facility Survey (NHFS) for 2021 here https://nepalhealthmag.com/nepal-health-facility-survey-2021-preliminary-report/#:~:text=The%202021%20Nepal%20Health%20Facility%20Survey%20%28NHFS%29%20is,those%20services%2C%20and%20the%20quality%20of%20client%20services. This may have been released after the manuscript was submitted and may show a different picture of the state of structural readiness in these facilities.

Based on the above, I recommend that the paper be published with minor changes to the title showing the time period E.g., Health system readiness (structural quality) of HFs for routine maternal and newborn health services in Nepal: Evidence from nationally representative health facility survey conducted between 2009 and 2014 (Adjusted to show time frame of survey report). Alternatively, the dataset used to arrive at the conclusions in the paper can be updated to more recent report data.

Reviewer #3: I congratulate the authors for this assessment of the availability and protection of services in maternal and neonatal health care.

However, the results indicate that private health units have quality structures to offer better care to mothers and newborns than public health units in remote communities, which I think is to be expected. It would be important for this evaluation to be carried out among the public sector health units.

Reviewer #4: The manuscript title: “Health system readiness (structural quality) of HFs for routine maternal and newborn health services in Nepal: Evidence from nationally representative health facility survey.”

This manuscript seems scientifically well written. The study used nationally representative data collected in 2015. However, the claim made by the manuscript is relevant to the present context of Nepal. After the federal system, local governments are establishing community-level health facilities, and the recommendations made by the manuscript will be applicable for maintaining quality MNH service.

The manuscript seems fit for the journal; however, I do have some minor comments in the following sections:

Please insert line number in your text (there is no line number after page 13)

Methods/study variables:

1) Line 189:

All the independent variables are self-explanatory; however, I think it is better to explain bit or refer about mechanism of quality assurance (independent variables)

2) Line 193 to 195: “This study has two outcome variables: Structural quality of HFs for i) first ANC visit 194 (poor, better), ii) delivery and postnatal services (poor, better). Structural quality of 195 HFs for the first ANC visit, and delivery and PNC services”

-I think we can minimize duplication.

Results:

3) Line 258: “More than three in four (76/9%)”

-I think it is 90% (Table 1)

4) Table 1:

- I think author need to write abbreviation in the legend for the readability

5) Page 14: “Distribution HFs with better structural quality for MNH services”

-“of” is missing

6) Page 14 second paragraph: “On the other hand, the HFs of the Mountain region and Karnali province (12%) had poor structural quality for delivery and postnatal services compared to province one (38%)”

- I think it is better to remove the "mountain region" as it is not study variable (for this manuscript).

7)For table 4 and 3

- Re-arrange the table number (4 came before 3)

- Please make consistency in upper/lower case while writing abbreviations in legend.QA: Quality assurance; HF: health facility, PHCCs: primary health care centers; HPs: health posts; ANC: Antenatal care; PNC: postnatal care

Reviewer #5: I have well read your work which i found very interesting and original. However, i have some questions:

1) in the methods section, you said WHO's SARA manual was the one you used to assess the structural quality of health facilities to provide ANC and delivery and post-natal services. However, i saw for ANC serices, 21 items that you used whereas in the manual, there are only 06 items. The same observation has been made for the delivery and post-natal services. I understand that you have contextualized the items since you added and removed some items. Please could you provide clear references and deep explanation on the choice of items you added to the list and those you removed from SARA's items?

2) As for the readiness, SARA's manual has presented 04 domains (staff and guidelines, equipment, diagnostic and medicines and commodities). I have understood that you added general rediness as domain and did not include the diagnostic domain. Could you clearly explain that?

3) in the results, the way you presented your tables especially tables 2 and 3 is not easily understandable by readers. Please, could you present the tables consistency?

4) you have calculated scores to define the structural quality of health facilities using availabity score and readiness score for each service. Based on the SARA manual, the way to calculate scores depends on the domains and items in each domain. Please, could you clearly explain the way you calculated facility readiness score based on the domains? Also, try to explain it in your methods section.

6. PLOS authors have the option to publish the peer review history of their article (what does this mean?). If published, this will include your full peer review and any attached files.

**Do you want your identity to be public for this peer review?** For information about this choice, including consent withdrawal, please see our Privacy Policy.

Reviewer #1: No

Reviewer #2: No

Reviewer #3: **Yes: **Cristolde Atanasio Salomao

Reviewer #4: **Yes: **Prasant Vikram Shahi

Reviewer #5: No

---

## [Editor Report · Decision Letter 1]

26 Oct 2022

Assessment of health system readiness for routine maternal and newborn health services in Nepal: Analysis of a nationally representative health facility survey, 2015

PGPH-D-22-00510R1

Dear Dr Khatri

We are pleased to inform you that your manuscript 'Assessment of health system readiness for routine maternal and newborn health services in Nepal: Analysis of a nationally representative health facility survey, 2015' has been provisionally accepted for publication in PLOS Global Public Health.

Best regards,

Augustine D. Asante

Academic Editor